# Mechanical ventilation for COVID-19: Outcomes following discharge from inpatient treatment

Mark J. Butler[1]*, Jennie H. Best[2], Shalini V. Mohan[2], Jennifer A. Jonas[1], Lindsay Arader[1,3], Jackson Yeh[1]

1 Institute of Health System Science, Feinstein Institutes for Medical Research, Northwell Health, New York, NY, United States of America, 2 Genentech Inc., South San Francisco, CA, United States of America, 3 St. John's University, Jamaica, NY, United States of America

* markbutler@northwell.edu

## Abstract

Though mechanical ventilation (MV) is used to treat patients with severe coronavirus disease 2019 (COVID-19), little is known about the long-term health implications of this treatment. Our objective was to determine the association between MV for treatment of COVID-19 and likelihood of hospital readmission, all-cause mortality, and reason for readmission. This study was a longitudinal observational design with electronic health record (EHR) data collected between 3/1/2020 and 1/31/2021. Participants included 17,652 patients hospitalized for COVID-19 during this period who were followed through 6/30/2021. The primary outcome was readmission to inpatient care following discharge. Secondary outcomes included all-cause mortality and reason for readmission. Rates of readmission and mortality were compared between ventilated and non-ventilated patients using Cox proportional hazards regression models. Differences in reasons for readmission by MV status were compared using multinomial logistic regression. Patient characteristics and measures of illness severity were balanced between those who were mechanically ventilated and those who were not utilizing 1-to-1 propensity score matching. The sample had a median age of 63 and was 47.1% female. There were 1,131 (6.4%) patients who required MV during their initial hospitalization. Rates (32.1% versus 9.9%) and hazard of readmission were greater for patients requiring MV in the propensity score–matched samples [hazard ratio (95% confidence interval) = 3.34 (2.72–4.10)]. Rates (15.3% versus 3.4%) and hazard [hazard ratio (95% confidence interval) = 3.12 (2.32–4.20)] of all-cause mortality were also associated with MV status. Ventilated patients were more likely to be readmitted for reasons which were classified as COVID-19, infectious diseases, and respiratory diagnoses compared to non-ventilated patients. Mechanical ventilation is a necessary treatment for severely ill patients. However, it may be associated with adverse outcomes including hospital readmission and death. More intense post-discharge monitoring may be warranted to decrease this associational finding.

**Data Availability Statement:** The current study looks at outcomes for patients who received mechanical ventilation (MV) for COVID-19 illness relative to a matched cohort of patients who did not. To comply with Safe Harbor de-identification

standards, some variables are omitted from the data posted on Open Science. The goal for removing this information is to prevent disclosure of personal health information (PHI). If you would like access to the full data, please contact Dr. Mark Butler (Study Primary Author, Institute of Health System Science, Northwell Health), Challace Pahlevan-Ibrekic (Director, Regulatory Affairs, Institute of Health System Science, Northwell Health), and Suzanne Ardito (Project Manager, Regulatory Affairs, Institute of Health System Science (IHSS), Northwell Health). Data requests can be made via email to markbutler@northwell. edu, cpahlevanibr@northwell.edu, and SArdito@northwell.edu. Data requests will be reviewed by the regulatory team and access to full data will be granted following Institutional Review Board (IRB) approval, as applicable, and completion of a data use and sharing agreement with Northwell Health.

**Funding:** JY received funding from the National Institute of Aging, grant R24AG064191 (https:// www.nia.nih.gov/). The funders had no role in study design, data collection and analysis, decision to publish, or preparation of the manuscript.

**Competing interests:** The authors have declared that no competing interests exist.

## Introduction

Patients with coronavirus disease 2019 (COVID-19) often suffer severe symptoms—from viral pneumonia to respiratory distress [1, 2]. This respiratory distress can lead to alveolar damage and fibrosis in the lungs [1], which reduces oxygen saturation in the blood [3]. COVID-19–related respiratory distress has been associated with higher rates of mortality and intensive care unit (ICU) admission than respiratory distress associated with other illnesses [1–3]. The recommended treatment for patients suffering from severe respiratory distress due to COVID-19 is mechanical ventilation (MV) [3–6], which provides oxygen to these critically ill patients and removes carbon dioxide from the blood [7]. The use of MV treatment for severe COVID-19 is common, with incidence ranging from 12.2% [8] to 33.1% [9] among inpatients in the New York City region. In patients with severe COVID-19, MV can be a life-saving therapy. However, MV is an invasive treatment that can also produce lung injury without careful monitoring [10, 11], leading researchers to debate when escalation to MV for COVID-19 illness and investigation of the long-term consequences of MV for COVID-19 is appropriate [12, 13].

Studies have examined the outcomes and correlates of MV treatment for COVID-19 during inpatient hospitalization [6, 14, 15] and the outcomes of COVID-19 following discharge; however, studies have not compared these outcomes based on MV status [8]. Little is known about the outcomes following discharge on the subset of COVID-19 patients who received MV, including whether they are at greater risk for readmission or for mortality compared with non-MV patients. Small-scale studies have followed patients with severe COVID-19 illness treated with MV [16], but no large-scale observational studies have examined longer-term outcomes in these patients. Given that prior studies examining outcomes following MV for illnesses other than COVID-19 have shown that MV patients may be at higher risk for hospital readmission and all-cause mortality [17, 18]. studying the outcomes for this subset of COVID-19 patients is particularly important. Understanding the outcomes following MV treatment is especially important considering that the ventilation treatment guidelines for COVID-19 were slow to develop during the early stages of the pandemic and there were high levels of uncertainty and confusion in how MV should be implemented among patients hospitalized with COVID-19 [19, 20].

This study examines whether MV treatment for COVID-19 illness is associated with readmission to inpatient treatment after the patient's initial hospitalization. It also examines whether MV is associated with higher levels of all-cause mortality. Finally, among patients who are readmitted to the hospital, it examines whether the reasons for readmission differ between patients who were treated with MV and patients who were not. The goal was to determine whether outcomes differed between patients with COVID-19 who received MV compared to those who did not while utilizing statistical methods to account for differences in patient demographics, comorbidity, medication treatment, and illness severity. These data will enable physicians and researchers to better understand how MV treatment uniquely contributed to patient outcomes among patients hospitalized with severe COVID-19 illness.

## Methods

### Study design and participants

This was a longitudinal observational study using electronic health record (EHR) data from the Northwell Health system. Northwell Health is a healthcare system that comprises 23 hospitals/medical facilities serving New York City, Long Island, and the surrounding area. This region was one of the epicenters of the COVID-19 pandemic in the United States [9].

The sample was composed of 17,562 patients hospitalized for COVID-19 between March 1, 2020, and January 31, 2021. Patient follow-up was conducted through June 30, 2021. COVID-

19 illness was defined as a positive polymerase chain reaction (PCR) test or by a diagnosis of COVID-19 in the patient's clinical chart. Patients <18 years of age and patients who died during their initial inpatient admission were excluded from the analysis. Patients or the public were not involved in the design, conduct, reporting, or dissemination plans of our research. STROBE reporting guidelines were used in this study [21]. The Northwell Health institutional review board approved this observational analysis as minimal-risk research using data collected for routine clinical practice and waived the requirement for informed consent.

## Mechanical ventilation status

Patients who received MV at any point during their initial inpatient treatment for COVID-19 were defined as "ventilated." Patients who were treated and discharged from their initial hospitalization without receiving MV were defined as "non-ventilated." For patients who were discharged from inpatient care and returned to inpatient care within 24 hours, the return to inpatient care was classified as being part of the initial hospitalization.

## Outcomes

Readmission to inpatient care was the primary outcome for the study. Readmission was defined as any inpatient admission occurring more than 24 hours after discharge from the initial hospitalization. Returning to inpatient care 24 hours or less following discharge was classified as a change in level of care rather than a discharge followed by a readmission.

Secondary outcomes included all-cause mortality and reason for readmission. All-cause mortality was defined using date and time of death recorded in the Northwell Health EHR. Reasons for readmission were defined based on the International Classification of Diseases (ICD)-10 codes recorded at the start of the second admission. Patient diagnoses were grouped together by category. The categories were defined as follows: COVID-19 (by provider diagnosis); cardiovascular and blood diseases; respiratory diseases; infectious diseases; endocrine disorders; mental/psychological diagnoses; nervous system disorders; disorders of the eyes, ears, and skin; digestive disorders; muscular disorders; genitourinary diagnoses; pregnancy-related diagnoses; birth-related diagnoses; abnormal symptoms/lab values; injuries; and other diagnoses. Details on categorization of reasons for readmission and associated ICD-10 codes can be found in S1 Table in S1 File.

## Potential confounding variables

Potentially confounding variables that might relate to our primary and secondary outcomes include the month of initial admission, patient demographics and comorbidities, treatment during the initial hospitalization, anthropometrics from the initial admission, and laboratory values approximating the severity of illness at presentation from the initial admission. Patient treatment and outcomes for COVID-19 may also differ based on the stage of the pandemic in which the patient became ill with COVID-19 and received treatment [22]. Providers who treated patients with COVID-19 earlier in the pandemic faced a scarcity of treatment resources [23, 24] and uncertainty about how to treat COVID-19 illness [25]. Therefore, the month in which a patient was initially hospitalized was a potential confounding variable.

Patient demographic variables used in the study included age, sex, race, ethnicity, and insurance status. Patient comorbidities examined included smoking status, pulmonary diseases, cardiovascular diseases, renal diseases, cancer, dementia, and immunodeficiency. Details of the patient's initial inpatient admission for treatment of COVID-19 included the length of stay and classes of medications the patient was treated with, including antiviral, anticoagulant, corticosteroid, interleukin-1 (IL1)- inhibitors, and interleukin-6 (IL6)-inhibitors. Length of

stay was defined as the duration in days between the patient's admission date and discharge date. The medications identified were based on broad classes that have been used as therapeutic agents for COVID-19 [26–30]. Medication classes were coded and verified by two trained physicians.

Patient anthropometric measures included height, weight, systolic blood pressure, diastolic blood pressure, and oxygen saturation (SpO2) measured at time of admission. If the first anthropometric measure was found to be invalid (e.g., a diastolic blood pressure of 3mmHg), the closest valid measure collected following admission was used. Laboratory values included ferritin, C-reactive protein, D-Dimer, creatinine, and others. Laboratory values treated as potential confounders in the study included ferritin, C-reactive protein, D-Dimer, creatinine, lymphocyte count, neutrophil count, lactate dehydrogenase (LDH), sodium, potassium, albumin, white blood cell count, platelet count, international normalized ratio (INR), procalcitonin, troponin, aspartate aminotransferase (AST), and alanine aminotransferase (ALT). Anthropometric and laboratory measures used were the first values collected following inpatient admission. All laboratory values utilized in the current study were selected because they have been identified as markers of COVID-19 illness severity [31–39] and/or were found to be associated with outcomes included in the study (i.e., readmission to inpatient care and mortality) [14, 40–48].

## Statistical analysis

### Descriptive analyses

Characteristics of the sample and potential confounding variables were reported for the full sample and by MV status using median and interquartile range (IQR) for continuous variables and frequencies and percentages for categorical variables. Comparisons of characteristics by MV status were conducted using Kruskal-Wallis tests for continuous variables and Pearson chi-squared tests for categorical variables. Absolute standardized differences (ASDs) were calculated between the MV and non-MV groups for all variables.

Many patients were missing data for anthropometric or lab values. To account for missing values, these variables were categorized based on clinical significance while retaining "missing" as a category. Variable categories were based on established clinical guidelines (e.g., World Health Organization categories for body mass index [BMI]) [49] or prior research linking lab values to outcomes for COVID-19 (e.g., ferritin greater than 800 ng/dl [32] and lactate dehydrogenase [LDH] greater than 255 U/L) [50].

### Propensity score matching

Given that MV patients were not likely to have equivalent levels of potential confounding variables to the non-MV patients, statistical measures were used to balance potential confounders between groups. One-to-one propensity score matching [51] was used to create a matched sample of MV and non-MV patients.

Balance of potential confounders between the MV and non-MV samples was examined using ASDs. Potential confounding variables were determined to be balanced between the MV and non-MV samples if the ASD between the two groups was less than 0.10. Samples were matched on all potential confounders using the package "MatchIt" [52] in R Version 4.1.0 [53]. Potential confounders that remained unbalanced between MV and non-MV groups (defined by ASD greater than or equal to 0.10) after matching were utilized as covariates in analyses.

One-to-one propensity score matching was also conducted in the sample that was readmitted to inpatient care to balance potential confounders in this sample of individuals. Readmitted

MV patients were matched with a cohort of individuals who were readmitted but did not receive MV using propensity scores. As with the primary matched sample, ASDs were used to check for confounder balance in the matched readmitted sample. Variables with ASDs greater than or equal to 0.10 were also utilized as covariates in these analyses.

### Analyses for the primary outcome: Readmission to inpatient care

Frequencies and percentages for all outcomes were reported overall and by MV status. Associations between readmission and MV status were conducted using Cox proportional hazards regression. For these models, the time-to-event was defined as the difference in days between the patient's initial discharge date and date of readmission. For patients who were not readmitted, time to censoring was defined as the difference between the patient's initial discharge and the date of the last day of follow-up (June 30, 2021).

### Analyses for the secondary outcomes: All-cause mortality and reason for readmission

Cox proportional hazards regression models were used to determine relative hazard of mortality in the MV group compared to the non-MV group. As with the analyses for the primary outcome, Cox proportional hazards regressions were conducted in the full sample and in the 1-to-1 propensity-matched samples.

Comparisons of reason for readmission between the MV and non-MV groups were conducted using multinomial logistic regression. To increase interpretability of the regression model, the top-five most frequently occurring reasons for readmission (abnormal symptoms and lab values, COVID-19, respiratory diagnoses, circulatory diagnoses, and infectious diseases) were compared to all other diagnoses as the reference group. For all outcomes, potential confounders that remained unbalanced between the MV and non-MV groups after propensity matching were utilized as covariates in the regression.

### Sensitivity analyses

Outcomes for COVID-19 have also been found to differ by patient race. In previous studies, individuals identifying as Black or African American showed more severe outcomes [54, 55], which was potentially explained by disparities in socioeconomic status [56, 57]. This study also examines the associations between MV and outcomes in the sample that identified as Black or African American. Finally, as corticosteroids have been found to be an effective treatment for COVID-19 illness [58], sensitivity analyses were conducted in the sample of patients treated with corticosteroid(s) during their initial inpatient admission. To reduce potential confounding in these sensitivity analyses, 1-to-1 propensity score matching was conducted for each sample. For all sensitivity analyses, patient characteristics were examined for the full sample and propensity-matched sample and ASDs were reported between the MV and non-MV groups. Cox proportional hazards regression analyses were used to compare hazard of readmission and all-cause mortality in the MV group relative to the non-MV group. Due to small sample sizes, comparisons of reasons for readmission between MV and non-MV groups were not conducted in sensitivity analyses.

## Results

### Descriptive analyses

The median age of the full sample was 63 [IQR = 25] years, 47.1% of the sample was female (8,280/17,652), and 19.4% of the sample identifying as Hispanic/Latinx (3,411/17,652). Within this sample, 6.4% of individuals were ventilated at some point during their initial

hospitalization for COVID-19 (1,131/17,652). The MV sample had a median age of 63 (IQR = 19) years, was 35.5% female (402/1,131), and was 22.5% Hispanic/Latinx (254/1,131). Full sample characteristics can be found in Table 1. Sample characteristics prior to and following propensity score matching are presented in S2 and S3 Tables in S1 File. Length of stay was the only potential confounding variable that remained unbalanced after propensity score matching (ASD = 0.37) and was utilized as a covariate in regression analyses.

## Primary outcome

**Readmission to inpatient care.** Individuals readmitted to inpatient care for COVID-19 following their initial discharge made up 11.4% of the total sample (1,994/17,562). Of the patients who received MV during their initial hospitalization, 32.1% were readmitted (362/1,131). Rates of readmission were lower among those who did not receive MV during initial hospitalization (9.9%; 1,632/16,431). Frequency of readmission by MV status and regression results are presented in Table 2.

Fig 1 shows the Kaplan-Meier curve for readmission over time in the full and propensity-matched samples. This analysis suggests that most readmissions occurred within the first 100 days following discharge in both the MV and non-MV samples. Cox proportional hazards regression model results for readmission showed that MV was associated with an increased hazard of readmission over time in the full sample [(HR [95% CI] = 3.60 [3.22 to 4.04]; p < .001), in the propensity score–matched sample (HR [95% CI] = 3.34 [2.72 to 4.10]; p < .001), and in the propensity-matched sample adjusted for length of stay (HR [95% CI] = 3.67 [2.99 to 4.53]; p < .001).

## Secondary outcomes

**All-cause mortality.** In the full sample, 4.2% of patients died after discharge (735/17,562), corresponding to 15.3% of the MV group (173/1,131) and 3.4% of the non-MV group (562/16,431). Frequency of death by MV status and regression results are presented in Table 2.

Fig 1 shows the Kaplan-Meier curve for mortality over time in the full and propensity-matched samples. This analysis suggests that most cases of all-cause mortality occurred within the first 100 days following discharge in both the MV and non-MV samples. The Cox proportional hazards regression model showed that, for all-cause mortality, MV was associated with an increased hazard of mortality over time in the full sample (HR [95% CI] = 4.76 [4.01 to 5.64]; p < .001), in the propensity score–matched sample (HR [95% CI] = 3.12 [2.32 to 4.20]; p < .001), and in the matched sample with adjustment for length of stay (HR [95% CI] = 3.79 [2.82 to 5.10]; p < .001).

## Reason for readmission

Of the 1,994 patients who were readmitted, 1,895 had a single ICD-10 code recorded at readmission. 86 patients had two diagnosis codes and 20 patients had three diagnosis codes. For patients with multiple diagnosis codes, COVID-19 (corresponding to ICD-10 codes of U07.1 or J12.82 at admission) was assigned first. If no COVID-19 diagnosis was present, other diagnoses were assigned with circulatory, respiratory, and blood diagnoses assigned first. All additional diagnoses were categorized in alphabetical order of ICD-10 codes (e.g., "A41.9" was assigned before "R06.02"). The list of diagnoses corresponding to each category of reason for readmission can be found in S1 Table in S1 File. Using these categorizations, we found that most participants were readmitted for COVID-19, abnormal symptoms or lab values, circulatory issues, infectious diseases, or respiratory issues. The most commonly occurring ICD-10 codes for patients in these categories are presented in S4 Table in S1 File. Characteristics of the

**Table 1. Patient characteristics, comorbidity, visit details, anthropometric/laboratory values by mechanical ventilation status.**

| | | Total (n = 17,562) | Did Not Receive Mechanical Ventilation (n = 16,431) | Received Mechanical Ventilation (n = 1,131) | Comparison p-value ‡ |
|---|---|---|---|---|---|
| **Month of Initial Admission** | March, 2020 | 2,984 (17.0%) | 2,574 (15.7%) | 410 (36.3%) | < .001 |
| | April, 2020 | 5,614 (32.0%) | 5,252 (32.0%) | 362 (32.0%) | |
| | May, 2020 | 1,184 (6.7%) | 1,142 (7.0%) | 42 (3.7%) | |
| | June, 2020 | 429 (2.4%) | 413 (2.5%) | 16 (1.4%) | |
| | July, 2020 | 307 (1.7%) | 295 (1.8%) | 12 (1.1%) | |
| | August, 2020 | 227 (1.3%) | 217 (1.3%) | 10 (0.8%) | |
| | September, 2020 | 251 (1.4%) | 238 (1.4%) | 13 (1.1%) | |
| | October, 2020 | 415 (2.4%) | 399 (2.4%) | 16 (1.4%) | |
| | November, 2020 | 960 (5.5%) | 919 (5.6%) | 41 (3.6%) | |
| | December, 2020 | 2,221 (12.6%) | 2,138 (13.0%) | 83 (7.3%) | |
| | January, 2021 | 2,970 (16.9%) | 2,844 (17.3%) | 126 (11.1%) | |
| **Demographics** | | | | | |
| **Age; Median (IQR)** | | 63 (25) | 63 (26) | 63 (19) | .101 |
| **Sex** | Female | 8,280 (47.1%) | 7,878 (47.9%) | 402 (35.5%) | < .001 |
| | Male | 9,282 (52.9%) | 8,553 (52.1%) | 729 (64.5%) | |
| **Race** | White | 7,597 (43.3%) | 7,148 (43.5%) | 449 (39.7%) | < .001 |
| | Black | 3,231 (18.4%) | 3,056 (18.6%) | 175 (15.5%) | |
| | Asian | 1,479 (8.4%) | 1,367 (8.3%) | 112 (9.9%) | |
| | Other/ Multiracial | 4,505 (25.7%) | 4,172 (25.4%) | 333 (29.4%) | |
| | Unknown | 750 (4.3%) | 688 (4.2%) | 62 (5.5%) | |
| **Ethnicity** | Hispanic/Latinx | 3,411 (19.4%) | 3,157 (19.2%) | 254 (22.5%) | < .001 |
| | Non-Hispanic | 13,164 (75.0%) | 12,373 (75.3%) | 791 (69.9%) | |
| | Other/ Unknown | 987 (5.6%) | 901 (5.5%) | 86 (7.6%) | |
| **Insurance** | Commercial | 6,156 (35.1%) | 5,736 (34.9%) | 420 (37.1%) | .033 |
| | Medicare | 7,185 (40.9%) | 6,769 (41.2%) | 416 (36.8%) | |
| | Medicaid | 3,809 (21.7%) | 3,537 (21.5%) | 272 (24.0%) | |
| | Self-Pay | 114 (0.6%) | 109 (0.7%) | 5 (0.4%) | |
| | Other | 298 (1.7%) | 280 (1.7%) | 18 (1.6%) | |
| **Comorbidity** | | | | | |
| **Smoking Status** | Current | 435 (2.5%) | 417 (2.5%) | 18 (1.6%) | < .001 |
| | Former | 1,975 (11.2%) | 1,859 (11.3%) | 116 (10.3%) | |
| | Never | 12,990 (74.0%) | 12,291 (74.8%) | 699 (61.8%) | |
| | Unknown | 1,938 (11.0%) | 1,664 (10.1%) | 274 (24.2%) | |
| | Missing | 224 (1.3%) | 200 (1.2%) | 24 (2.1%) | |
| **Asthma** | | 1,235 (7.0%) | 1,144 (7.0%) | 91 (8.0%) | .187 |
| **COPD** | | 1,022 (5.8%) | 958 (5.8%) | 64 (5.7%) | .863 |
| **Obstructive Sleep Apnea** | | 563 (3.2%) | 509 (3.1%) | 54 (4.8%) | .003 |
| **Hypertension** | | 8,682 (49.4%) | 8,066 (49.1%) | 616 (54.5%) | .001 |
| **Myocardial Infarction** | | 280 (1.6%) | 239 (1.5%) | 41 (3.6%) | < .001 |
| **Heart Failure** | | 1,411 (8.0%) | 1,297 (7.9%) | 114 (10.1%) | .010 |
| **Stroke / Ischemic Disease** | | 242 (1.4%) | 220 (1.3%) | 22 (1.9%) | .119 |
| **Aortic Aneurysm** | | 57 (0.3%) | 55 (0.3%) | 2 (0.2%) | .527 |
| **CVD (all)** | | 1,607 (9.2%) | 1,464 (8.9%) | 143 (12.6%) | < .001 |
| **Diabetes Mellitus** | | 792 (4.5%) | 730 (4.4%) | 62 (5.5%) | .120 |
| **CKD** | | 1,826 (10.4%) | 1,701 (10.4%) | 125 (11.1%) | .487 |

*(Continued)*

**Table 1.** (Continued)

| | | Total (n = 17,562) | Did Not Receive Mechanical Ventilation (n = 16,431) | Received Mechanical Ventilation (n = 1,131) | Comparison p-value ‡ |
|---|---|---|---|---|---|
| **Cancer** | | 1,442 (8.2%) | 1,359 (8.3%) | 83 (7.3%) | .294 |
| **Dementia** | | 1,224 (7.0%) | 1,186 (7.2%) | 38 (3.4%) | < .001 |
| **Immunodeficiency** | | 88 (0.5%) | 80 (0.5%) | 8 (0.7%) | .425 |
| **Visit Details** | | | | | |
| **Length of Stay; Median (IQR)** | | 5 (7) | 5 (6) | 19 (27) | < .001 |
| **Antiviral Treatment** | | 9,955 (56.7%) | 9,032 (55.0%) | 923 (81.6%) | < .001 |
| **Anticoagulant Treatment** | | 15,887 (90.5%) | 14,766 (89.9%) | 1,121 (99.1%) | < .001 |
| **Corticosteroid Treatment** | | 8,651 (49.3%) | 7,760 (47.2%) | 891 (78.8%) | < .001 |
| **IL-1 Inhibitor Treatment** | | 673 (3.8%) | 520 (3.2%) | 153 (13.5%) | < .001 |
| **IL-6 Inhibitor Treatment** | | 1,081 (6.2%) | 793 (4.8%) | 288 (25.5%) | < .001 |
| **Anthropometrics and Laboratory Values** | | | | | |
| **Height; Median (IQR)** | | 167.6 (15.2) | 168.0 (15.2) | 168.0 (12.7) | < .001 |
| **Weight; Median (IQR)** | | 76.2 (26.3) | 75.7 (26.3) | 80.0 (25.5) | < .001 |
| **BMI; Median (IQR)** | | 26.9 (8.2) | 26.8 (8.2) | 27.5 (9.0) | < .001 |
| **Systolic BP; Median (IQR)** | | 130.0 (30.0) | 130.0 (29.5) | 130 (31.0) | .328 |
| **Diastolic BP; Median (IQR)** | | 76.0 (17.0) | 76.0 (16.0) | 75.0 (18.0) | .001 |
| **SpO2; Median (IQR)** | | 96 (5) | 96 (5) | 93 (11) | < .001 |
| **SpO2** | ≤ 94% | 6,473 (36.9%) | 5,809 (35.4%) | 664 (58.7%) | < .001 |
| | > 94% | 11,041 (62.9%) | 10,574 (64.4%) | 467 (41.3%) | |
| **Ferritin; Median (IQR)** | | 619.0 (875.5) | 601.0 (856.0) | 837.0 (1,167.0) | < .001 |
| Ferritin | ≤ 800 ng/dl | 8,264 (47.1%) | 7,756 (47.2%) | 508 (44.9%) | < .001 |
| | > 800 ng/dl | 5,419 (30.9%) | 4,870 (29.6%) | 549 (48.5%) | |
| C-reactive protein; Median (IQR) | | 9.63 (16.36) | 9.16 (15.70) | 16.00 (19.30) | < .001 |
| **C-reactive protein** | ≤ 30 mg/dl | 11,139 (63.4%) | 10,337 (62.9%) | 802 (70.9%) | < .001 |
| | > 30 mg/dl | 2,250 (12.8%) | 2,015 (12.3%) | 235 (20.8%) | |
| **D-Dimer; Median (IQR)** | | 417 (558) | 405 (518) | 620 (1,588) | < .001 |
| **D-Dimer** | ≤ 1000 ng/ml | 10,485 (59.7%) | 9,709 (59.1%) | 776 (68.6%) | < .001 |
| | > 1000 ng/ml | 1,463 (8.3%) | 1,218 (7.4%) | 245 (21.7%) | |
| **Creatinine; Median (IQR)** | | 1.00 (0.55) | 0.99 (0.54) | 1.05 (0.60) | < .001 |
| **Lymphocyte count; Median (IQR)** | | 0.99 (0.76) | 1.00 (0.77) | 0.85 (0.63) | < .001 |
| **Neutrophil count; Median (IQR)** | | 5.53 (4.08) | 5.46 (3.99) | 6.68 (5.63) | < .001 |
| **Lactate Dehydrogenase; Median (IQR)** | | 350 (214) | 340 (202) | 478 (295) | < .001 |
| **Sodium; Median (IQR)** | | 136 (5) | 137 (5) | 135 (6) | < .001 |
| **Potassium; Median (IQR)** | | 4.0 (0.7) | 4.0 (0.7) | 4.1 (0.8) | .097 |
| **Albumin; Median (IQR)** | | 3.5 (0.8) | 3.5 (0.8) | 3.4 (0.9) | < .001 |
| **White Blood Cell count; Median (IQR)** | | 7.40 (4.51) | 7.35 (4.45) | 8.48 (5.95) | < .001 |
| **Platelet Count; Median (IQR)** | | 214 (113) | 214 (113) | 213 (120) | .461 |
| **International Normalized Ratio; Median (IQR)** | | 1.16 (0.21) | 1.15 (0.20) | 1.20 (0.24) | < .001 |
| **Procalcitonin; Median (IQR)** | | 0.15 (0.27) | 0.15 (0.25) | 0.34 (0.85) | < .001 |
| **Troponin; Median (IQR)** | | 0.06 (0.12) | 0.05 (0.11) | 0.11 (0.29) | < .001 |
| **Aspartate aminotransferase; Median (IQR)** | | 38 (34) | 37 (33) | 54 (47) | < .001 |

(*Continued*)

**Table 1.** (Continued)

| | Total (n = 17,562) | Did Not Receive Mechanical Ventilation (n = 16,431) | Received Mechanical Ventilation (n = 1,131) | Comparison p-value ‡ |
|---|---|---|---|---|
| **Alanine aminotransferase: Median (IQR)** | 30 (32) | 30 (31) | 37 (38) | < .001 |

All values above reported as frequency and percentages unless otherwise noted.

IQR = Interquartile Range; COPD = Chronic Obstructive Pulmonary Disease; CVD = Cardiovascular Disease; CKD = Chronic Kidney Disease; BMI = Body Mass Index; SpO2 = Oxygen Saturation

‡ Kruskal-Wallis tests and chi-squared tests used to generate p-values for continuous and categorical variables respectively

readmitted sample prior to and following propensity score matching are presented in S5 and S6 Tables in S1 File. Frequencies of reason for readmission by ventilation group are shown in Table 3. Frequencies of reason for readmission by ventilation group in the propensity-matched sample are shown in S7 Table in S1 File.

To increase interpretability of the regression model, several categories of reason for readmission were collapsed. The most frequently occurring categories (i.e., COVID-19, abnormal symptoms/labs, cardiovascular, respiratory, and infectious disease) were maintained. All other categories were collapsed into "other." The "other" group was set as the reference category for the multinomial logistic regression. Variables that were not balanced after propensity score matching (ASD greater than or equal to 0.10) were included as covariates. Results for the multinomial regressions are displayed in Table 4. Characteristics of the readmitted sample are presented prior to (S5 Table in S1 File) and after propensity score matching (S6 Table in S1 File).

**Table 2.** Analyses for primary outcome (readmission to hospital) and secondary outcome (all-cause mortality).

| Primary Outcome *Readmission to Hospital* | Category | Frequencies | | Cox PH Regression | |
|---|---|---|---|---|---|
| | | **Unadjusted Sample** | | | |
| Non-MV | Total (n = 17,562) | Readmitted (n = 1,994; 11.4%) | Hazard Ratio (95% CI) | Hazard Ratio ‡ (95% CI) | |
| | 16,431 (93.6%) | 1,632 (9.9%) | REF | REF | |
| MV | 1,131 (6.4%) | 362 (32.1%) | 3.60*** (3.22 to 4.04) | 4.13*** (3.63 to 4.72) | |
| | | **Propensity Score–Matched Sample** | | | |
| Non-MV | Total (n = 2,262) | Readmitted (n = 485; 21.4%) | Hazard Ratio (95% CI) | Hazard Ratio ‡ (95% CI) | |
| | 1,131 (50.0%) | 123 (10.9%) | REF | REF | |
| MV | 1,131 (50.0%) | 362 (32.1%) | 3.34*** (2.72 to 4.10) | 3.67*** (2.99 to 4.53) | |
| Secondary Outcome *All-cause Mortality* | | **Unadjusted Sample** | | | |
| Non-MV | Total (n = 17,562) | Mortality (n = 735; 4.2%) | Hazard Ratio (95% CI) | Hazard Ratio ‡ (95% CI) | |
| | 16,431 (93.6%) | 562 (3.4%) | REF | REF | |
| MV | 1,131 (6.4%) | 173 (15.3%) | 4.76*** (4.01 to 5.64) | 5.64*** (4.62 to 6.88) | |
| | | **Propensity Score–Matched Sample** | | | |
| Non-MV | Total (n = 2,262) | Mortality (n = 232; 10.3%) | Hazard Ratio (95% CI) | Hazard Ratio ‡ (95% CI) | |
| | 1,131 (50.0%) | 59 (5.2%) | REF | REF | |
| MV | 1,131 (50.0%) | 173 (15.3%) | 3.12*** (2.32 to 4.20) | 3.79*** (2.82 to 5.10) | |

*p < .05

**p < .01

***p < .001

‡ Covariate adjusted for length of stay

MV = Mechanical Ventilation

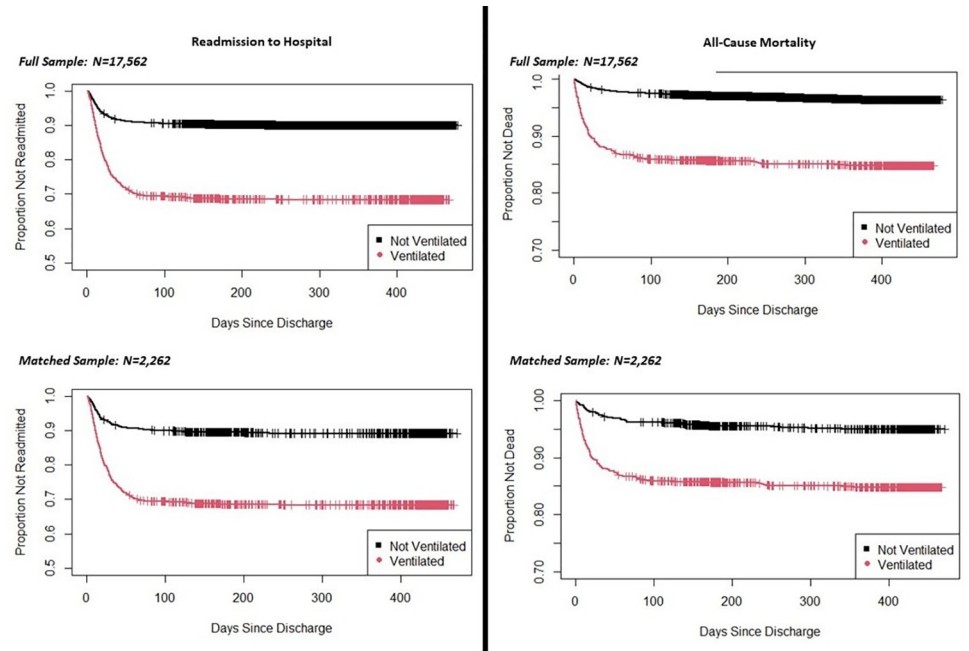

**Fig 1. Kaplan-Meier curve for readmission to hospital and all-cause mortality by treatment group.**

The multinomial logistic regression showed that individuals who received MV (compared to individuals who did not receive MV) had greater odds of being readmitted to inpatient care for COVID-19, infectious diseases, or respiratory issues relative to other diagnoses. These

**Table 3. Reasons for readmission by mechanical ventilation treatment status.**

| Reason for Readmission | Total (n = 1,994) | Did Not Receive Mechanical Ventilation (n = 1,632) | Received Mechanical Ventilation (n = 362) | Comparison p-value ‡ |
|---|---|---|---|---|
| Abnormal Symptoms & Labs | 529 (26.5%) | 465 (28.5%) | 64 (17.7%) | < .001 |
| COVID-19 | 523 (26.2%) | 379 (23.2%) | 144 (39.8%) | < .001 |
| Respiratory | 217 (10.9%) | 152 (9.3%) | 65 (18.0%) | < .001 |
| Circulatory Issues | 151 (7.6%) | 140 (8.6%) | 11 (3.0%) | < .001 |
| Infectious Disease | 99 (5.0%) | 68 (4.2%) | 31 (8.6%) | < .001 |
| Digestive | 75 (3.8%) | 69 (4.2%) | 6 (1.7%) | .030 |
| Blood Disease | 56 (2.8%) | 49 (3.0%) | 7 (1.9%) | .348 |
| Genitourinary | 54 (2.7%) | 53 (3.2%) | 1 (0.3%) | .003 |
| Mental | 54 (2.7%) | 53 (3.2%) | 1 (0.3%) | .003 |
| Endocrine | 47 (2.4%) | 42 (2.6%) | 5 (1.4%) | .246 |
| Injury | 40 (2.0%) | 35 (2.1%) | 5 (1.4%) | .465 |
| Other | 40 (2.0%) | 31 (1.9%) | 9 (2.5%) | .608 |
| Pregnancy | 30 (1.5%) | 30 (1.8%) | 0 (0.0%) | .018 |
| Nervous System | 28 (1.4%) | 20 (1.2%) | 8 (2.2%) | .232 |
| Muscular | 24 (1.3%) | 21 (1.3%) | 3 (0.8%) | .648 |
| Eyes, Ears, and Skin | 22 (1.1%) | 22 (1.3%) | 0 (0.0%) | .052 |
| Birth | 5 (0.3%) | 3 (0.2%) | 2 (0.6%) | .491 |

‡ Chi-squared tests used to generate p-values

**Table 4. Multinomial logistic regression for reason for readmission.**

| | Other | Abnormal Labs/ Symptoms | Circulatory | COVID-19 | Infectious Disease | Respiratory |
|---|---|---|---|---|---|---|
| | | | Readmitted Sample, N = 1,994 | | | |
| Non-MV | REF | REF | REF | REF | REF | REF |
| MV; OR (95% CI) | REF | 1.25 (0.84 to 1.87) | 0.72 (0.36 to 1.42) | 3.46*** (2.42 to 4.95) | 4.15*** (2.47 to 6.99) | 3.89*** (2.56 to 5.92) |
| | | | Readmitted Sample with Covariate Adjustment, N = 1,994 | | | |
| Non-MV | REF | REF | REF | REF | REF | REF |
| MV; OR (95% CI) | REF | 1.38 (0.84 to 2.28) | 0.64 (0.28 to 1.48) | 3.71*** (2.31 to 5.97) | 5.75*** (2.91 to 11.37) | 5.00*** (2.90 to 8.62) |
| | | | Readmitted Sample with 1-to-1 Propensity Matching, N = 724 | | | |
| Non-MV | REF | REF | REF | REF | REF | REF |
| MV; OR (95% CI) | REF | 0.87 (0.54 to 1.42) | 0.50 (0.23 to 1.08) | 2.36*** (1.50 to 3.71) | 3.08*** (1.50 to 6.32) | 2.20*** (1.29 to 3.75) |
| | | | Readmitted Sample with 1-to-1 Propensity Matching and Covariate Adjustment, N = 724 | | | |
| Non-MV | REF | REF | REF | REF | REF | REF |
| MV; OR (95% CI) | REF | 1.03 (0.58 to 1.83) | 0.50 (0.20 to 1.24) | 3.01*** (1.75 to 5.18) | 4.54*** (2.02 to 10.20) | 3.48*** (1.87 to 6.46) |

*p < .05

**p < .01

***p < .001

Other Diagnoses Include: Digestive, Blood disease, Genitourinary, Mental, Endocrine, Injury, Other, Pregnancy, Nervous System, Muscular, Eyes/Ears/Skin, and Birth

Covariates include: month of admission, age, race, insurance status, smoking status, CKD history, length of stay, corticosteroid treatment, IL-1 inhibitor treatment, IL-6 inhibitor treatment, BMI, diastolic blood pressure, oxygen saturation, ferritin, D-Dimer, creatinine, neutrophil, sodium, LDH, INR, and procalcitonin.

significant associations persisted across all four models. MV was not associated with increased odds of readmission for abnormal symptoms/labs or circulatory issues relative to other diagnoses.

## Sensitivity analyses

Sensitivity analyses were conducted in the sample that identified as Black/African American (N = 3,231) and the sample treated with corticosteroids during initial admission (N = 8,651). In each of these samples, MV was associated with increased odds of readmission and mortality.

**Race.** The first set of sensitivity analyses focused on the sample identifying as Black/African American. There were 3,231 patients who identified as Black/African American, 5.4% of which received MV (175/3,231) and 94.6% of which did not (3,056/3,231). Rates of MV were lower in patients identifying as Black/African American (5.4%) compared to the full sample (6.4%). S8a Table in S1 File shows the characteristics of the Black/African-American sample. Using 1-to-1 propensity score matching, a matched cohort was created in the Black/African-American sample; it comprised of 175 MV individuals and 175 non-MV individuals. Descriptive statistics for this matched sample are shown in S8b Table in S1 File. Variables that remained unbalanced in the matched sample included length of stay, month of admission, smoking status, insurance status, hypertension status, systolic blood pressure, diastolic blood pressure, SpO2, ferritin, c-reactive protein, and D-Dimer. These variables were used as covariates in adjusted models.

In the Black/African-American sample, of the 366 (11.3% of 3,231) patients who were readmitted after discharge, 56 (32.0% of 175 patients) were ventilated and 310 (10.1% of 3,056 patients) were not ventilated. In the propensity matched sample who identified as Black/African American, of the 78 (22.3% of 350) patients who were readmitted, 56 (32.0% of 175 patients) were ventilated and 22 (12.6% of 175 patients) were not. Cox proportional hazards

regression analyses showed that ventilation was associated with increased odds of readmission in the Black/African-American sample (HR [95% CI] = 3.54 [2.66 to 4.71; p < .001) and in the propensity score–matched Black/African-American sample (HR [95% CI] = 2.92 [1.78 to 4.78]; p < .001). Covariate adjusted models did not converge and as a result, estimates from these models should be interpreted cautiously. Full results is shown in S8c Table in S1 File.

In the Black/African-American sample, 5.4% of patients died after discharge (175/3,231), accounting for 16.6% of those who received MV (29/175) and 3.2% of those who did not (97/3,056). In the propensity matched sample who identified as Black/African American, 9.4% of patients died after discharge (33/350), accounting for 16.6% of those who received MV (29/175) and 2.3% of those who did not (4/175). Cox proportional hazards regression analyses showed that MV was associated with increased odds of all-cause mortality in the Black/African-American sample (HR [95% CI] = 5.62 [3.71 to 8.52]; p < .001) and in the propensity score–matched Black/African-American sample (HR [95% CI] = 7 .75 [2.72 to 22.04]; p < .001). Covariate adjusted models did not converge and as a result, estimates from these models should be interpreted cautiously. Full results from this sample are shown in S8c Table in S1 File. The sample of patients who identified as Black/African-American (N = 3,231) had comparable rates of readmission (11.3% versus 11.4%) and increased rates of all-cause mortality (5.4% versus 4.2%) compared to the full sample (N = 17,562). The most notable difference in the sample identifying as Black/African-American was that MV was associated with a much higher hazard of mortality across all levels of adjustment.

**Corticosteroid treatment.** The second set of sensitivity analyses focused on patients treated with a corticosteroid during their initial admission. Of the 8,651 patients treated with corticosteroids, 10.3% received MV (891/8,651) and 89.7% did not (7,760/8,651). Rates of MV were higher in the group treated with corticosteroids (10.3%) compared to the full sample (6.4%). S9a Table in S1 File shows characteristics of the sample treated with corticosteroids. Using 1-to-1 propensity score matching, a matched cohort was created in the corticosteroid treated sample that was comprised of 891 MV individuals and 891 non-MV individuals, creating a matched sample of 1,782 patients treated with a corticosteroid during their initial admission. Descriptive statistics for this sample are shown in S9b Table in S1 File. Variables that remained unbalanced in the matched sample included length of stay and race. These variables were used as covariates in adjusted models.

In the sample treated with a corticosteroid during initial admission, 10.3% of patients were readmitted (831/ 8,651), accounting for 32.1% of those who received MV (286/891) and 10.0% of those who did not (775/7,760). In the propensity matched sample who were treated with corticosteroids, 21.4% of patients were readmitted (382/1,782), accounting for 32.1% (286/891) of those who received MV and 10.8% of those who did not (96/891). Cox proportional hazards regression analyses showed that MV was associated with increased hazard of readmission in the corticosteroid-treated sample (HR [95% CI] = 3.57 [3.11 to 4.08]; p < .001), the corticosteroid-treated sample with covariate adjustment (HR [95% CI] = 4.48 [3.83 to 5.24]; p < .001), the propensity score–matched corticosteroid-treated sample (HR [95% CI] = 3.37 [2.68 to 4.25]; p < .001), and the propensity score–matched corticosteroid-treated sample with covariate adjustment (HR [95% CI] = 3.83 [3.03 to 4.85]; p < .001). Full information on the corticosteroid-treated sample is shown in S9c Table in S1 File.

In the corticosteroid-treated sample, 4.8% of patients died after discharge (419/8,651), accounting for 16.0% of those who received MV (143/891) and 3.6% of who did not (276/7,760). In the propensity matched sample that was treated with corticosteroids, 10.1% died after discharge (180/1,782), accounting for 16.0% of those who received MV (143/891) and 4.2% of those who did not (37/891). Cox proportion hazards regression analyses showed that MV was associated with increased hazard of all-cause mortality in the corticosteroid-treated

sample (HR [95% CI] = 4.69 [3.83 to 5.73]; p < .001), the corticosteroid-treated sample with covariate adjustment (HR [95% CI] = 6.39 [5.06 to 8.07]; p < .001), the propensity score–matched corticosteroid-treated sample (HR [95% CI] = 4.13 [2.88 to 5.93]; p < .001), and the propensity score–matched corticosteroid-treated sample with covariate adjustment (HR [95% CI] = 5.30 [3.68 to 7.63]; p < .001). Full information on the corticosteroid-treated sample is shown in S9c Table in S1 File.

## Discussion

Results of the current study demonstrate that individuals hospitalized with COVID-19 and treated with MV have a greater likelihood of adverse outcomes, including readmission to the hospital and all-cause mortality, following discharge from inpatient care than non-MV patients. Further, MV patients who were readmitted were more likely to be readmitted for COVID-19 illness, infectious diseases, and respiratory diagnoses than non-MV patients. This suggests that in addition to being more likely to be readmitted than non-MV patients, MV patients also have different presenting problems upon readmission.

Prior research has shown persistent and long-lasting physical and functional deficits in patients with severe COVID-19 following discharge from inpatient care [59] The current study expands upon that literature by identifying the especially vulnerable population of patients who received MV. Further, patients treated for MV who are readmitted to inpatient care were more likely to be readmitted for diagnoses relating to COVID-19, infectious diseases, and respiratory problems. These findings suggest that outpatient follow-up for ventilated patients should target symptoms relating to infection, respiration, and symptoms of COVID-19 to reduce their likelihood of future inpatient admissions. Sensitivity analyses also suggest that the associations between MV and readmission persist even in subsets of this population. Though results from these sub-samples must be interpreted cautiously due to smaller samples of patients receiving MV treatment, these sensitivity analyses suggest that the magnitude of the association between MV treatment for COVID-19 and readmission/mortality may vary by demographic group and due to interactions with other treatments (such as corticosteroids).

Prior research examining outcomes for non-COVID-19 related acute respiratory distress syndrome (ARDS) and acute respiratory failure (ARF) show high levels of readmission (18% to 53%) [60–62] and mortality (31% to 66%) [60, 61, 63] among patients receiving MV treatment. With these findings in mind, the rates of readmission (32.1%) and mortality (15.1%) for patients with COVID-19 who received MV treatment are comparable to or lower than what may be expected among patients with severe respiratory distress. However, these findings are novel because COVID-19 illness may not be entirely comparable to previous causes for ARDS. Firstly, there is controversy regarding the phenotypes underlying traditional ARDS and respiratory failure due to COVID-19 illness [19]. Secondly, the clinical presentation for respiratory failure in COVID-19 has been demonstrated to be more variable and non-uniform than what is traditionally seen in ARDS [19, 64]. With these facts in mind, comparing outcomes following MV treatment for COVID-19 illness and non-COVID-19 ARDS is difficult. Further, the current findings expand upon prior research, which primarily focuses on outcomes for MV treatment during hospitalization by following patients after discharge [14, 65–67].

The results of this study suggest that identifying solutions for MV-patients may be warranted. Fortunately, there are several possible solutions for addressing the problem that MV patients experience increased risk of readmission and mortality. One potential solution is to create follow-up and support programs for patients who were ventilated to ensure that they receive outpatient follow-up to reduce their likelihood of re-hospitalization and death. Another possible solution is to alter MV treatment to better suit the needs of critically ill

patients with COVID-19. Some work has already been done to evaluate the development of personalized medicine approaches to MV treatment [68–70], which may help tailor ventilation practices to individual patients and reduce adverse outcomes. As rates of vaccination increase, the importance of MV in COVID-19 treatment may decline [71]. This will reduce rates of MV and potential adverse outcomes following MV treatment. Regardless, this study shows that patients treated with MV for COVID-19 are at increased risk for adverse outcomes and need additional follow-up and specialized care after discharge from inpatient treatment.

## Strengths/Limitations

This study has several strengths. First, the analysis benefits from a large sample size of 17,562 patients with COVID-19 in the Northwell Health system. The size of the Northwell Health system allowed for access to a large amount of data from the early stages of the pandemic in the United States from March 2020 through the middle of 2021. Second, this study uses EHR data, which makes it possible to control for multiple important, potential confounding variables. This study was able to account for numerous factors that might influence rates of readmission, all-cause mortality, and reasons for readmission.

This study also has several limitations. Firstly, the analysis was conducted using data from one health system. Thus, patients outside of the Northwell Health network or New York region were not recorded in the dataset. Secondly, the definition of all-cause mortality in the analysis is based on hospital data. Thus, patients who died but did not have their death recorded in the medical record did not have this data captured. Finally, this study is observational. Given that a randomized controlled trial that assigns patients to an MV or non-MV group is not feasible, this analysis is subject to all of the limitations associated with observational studies. Specifically, patients who receive MV are more severely ill than patients who do not receive MV. Even in utilizing a propensity-matched control group to compare against the MV sample, it is not possible to fully account for underlying differences in disease severity between the two samples in our regression models. Thus, the associations demonstrated in this study may be due to the increased severity of illness of patients in the MV group rather than from the MV treatment itself. Further, it is difficult to identify the mechanisms of the association between MV and adverse outcomes shown in this sample. These associations could be due to factors associated with COVID-19, issues with the application of MV treatment, or other unknown factors.

## Conclusions

Findings from this study suggest that MV patients have a greater hazard of inpatient readmission and all-cause mortality compared to non-MV patients. Whether this is due to a difference in severity of illness (for which MV may be a proxy) or a consequence of MV itself, patients who are ventilated appear to be at greater risk for adverse outcomes following discharge. Regardless of the cause of association between adverse outcomes and MV treatment, the current study suggests that patients with COVID-19 illness who are treated with MV should be provided with additional support and follow-up after discharge from inpatient care.

## Supporting information

**S1 File.**
(DOCX)

## Acknowledgments

MB, JB, SM, JJ, and LA were responsible for drafting the manuscript. JY secured access to the data. MB, LA, and JY conducted statistical analyses. SM and JJ provided clinical expertise.

## Author Contributions

**Conceptualization:** Mark J. Butler, Jennie H. Best, Shalini V. Mohan, Jennifer A. Jonas, Jackson Yeh.

**Data curation:** Jennie H. Best, Shalini V. Mohan.

**Formal analysis:** Mark J. Butler, Jackson Yeh.

**Funding acquisition:** Jackson Yeh.

**Methodology:** Mark J. Butler, Jennie H. Best, Shalini V. Mohan, Jennifer A. Jonas, Lindsay Arader, Jackson Yeh.

**Resources:** Mark J. Butler.

**Writing – original draft:** Mark J. Butler.

**Writing – review & editing:** Mark J. Butler, Jennie H. Best, Shalini V. Mohan, Jennifer A. Jonas, Lindsay Arader, Jackson Yeh.

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
