## [Decision Letter · Decision Letter 0]

6 Jul 2022

PONE-D-22-12638Mechanical Ventilation for COVID-19: Outcomes Following Discharge from Inpatient TreatmentPLOS ONE

Dear Dr. Butler,

Thank you for submitting your manuscript to PLOS ONE. After careful consideration, we feel that it has merit but does not fully meet PLOS ONE’s publication criteria as it currently stands. Therefore, we invite you to submit a revised version of the manuscript that addresses the points raised during the review process.

This manuscript was peer-reviewed by three reviewers. Although the analysis methodology and the presentation of the result do not seem to have major problems, the reviewers have mentioned some major concerns in this manuscript regarding the scientific significance of this study, the interpretation and discussion of the results, and the data collection methodology. Therefore, the authors need to respond to all reviewers' comments. Especially please clarify the purpose and scientific significance in the introduction and the conclusions that can be drawn from the results of this study so that readers can understand them.

We look forward to receiving your revised manuscript.

Kind regards,

Masaki Tago, M.D., Ph.D.

Academic Editor

PLOS ONE

Journal Requirements:

3. Please include your tables as part of your main manuscript and remove the individual files. Please note that supplementary tables (should remain/ be uploaded) as separate "supporting information" files

Reviewers' comments:

Reviewer's Responses to Questions

**Comments to the Author**

1. Is the manuscript technically sound, and do the data support the conclusions?

Reviewer #1: Partly

Reviewer #2: Yes

Reviewer #3: Yes

2. Has the statistical analysis been performed appropriately and rigorously? 

Reviewer #1: Yes

Reviewer #2: Yes

Reviewer #3: Yes

3. Have the authors made all data underlying the findings in their manuscript fully available?

Reviewer #1: No

Reviewer #2: Yes

Reviewer #3: Yes

4. Is the manuscript presented in an intelligible fashion and written in standard English?

Reviewer #1: Yes

Reviewer #2: Yes

Reviewer #3: Yes

5. Review Comments to the Author

Reviewer #1: Thank you for submitting your research article entitled “Mechanical ventilation for COVID-19: outcomes following discharge from inpatient treatment”. Authors evaluated the long-term outcomes of severe COVID-19 patients with or without mechanical ventilation (MV), its impacts on the hospital readmission, all-cause mortality, and reason for readmission. I think this manuscript is very meaningful for all medical personnel who are involved in COVID-19 pandemic. However, there are some major concerns that should be addressed by the authors at this time.

<major comments="">

1. First of all, I think the topic of this study have little impact for the readers of this journal so that authors have not fully discussed in the DISCUSSION section. It is easy to imagine that the clinical prognosis of cases requiring mechanical ventilation is poor not only for other infections but also for COVID-19. Are there any differences in the characteristics or prognostic tendencies peculiar to patients with COVID-19?

2. In the INTRODUCTION section, authors described that MV can produce lung injury, which lead to poor long-term outcomes in this condition. Among the present study populations, authors should discuss whether these poor outcomes and high readmission rates are due to the effects of mechanical ventilation management, the complications of COVID-19, or other factors?

3. Authors described the limitation of data sampling in terms of readmission rate. Although I am not sure how many patients’ readmissions this healthcare system (Northwell Health) can cover, this limitation should be fully discussed since readmission rate is a primary outcome in this study. I would like to know how far the readmission ate in this study is from the real-world readmission rate.

4. Are there any standardized manual for MV management in these 23 medical facilities? If not, how did the physicians decide the indication of MV? Did you employ high-flow nasal cannula (HFNC) or non-invasive positive pressure ventilation (NPPV)? Moreover, are there any relationships between the duration of the intubation and readmission rate?

5. In the CONCLUSIONS, second paragraph [P15L11-P16L2] is not based on the data of this research. This is not conclusion of this study.

<minor comments="">

1. [P5L18] January 31, 2020 -> January 31, 2021

2. There are no “Table 1”, ”Table2” and “Table 3”.

3. For propensity score matching, please list the factors in order to make the adjustments.</minor></major>

Reviewer #2: Butler et al. reported the outcome of COVID-19 patients following discharge differentiating between those who needed mechanical ventilation and those without it.

As expected, patients who required MV were more susceptible to readmission and lower survival. The authors state that this feature is clearly recognized in ordinary ICU patients (not COVID), so the objective of this study should be clarified. Was their hypothesis that COVID patients behave differently (better or worse) than ordinary ICU patients? Then, the readers should be informed about how different the present results are compared with series of non-COVID patients. Nevertheless, even these comparisons are tricky because non-COVID severe ARF commonly affect patients with severe sepsis or severe comorbidities that explained most of the long-term outcome worsening.

Additional comments:

1.- Page 15, conclusions should be tailored by deleting the first 3 lines: “MV is an essential treatment ……... important”.

Also, the second paragraph about possible solutions must be moved to the end of the discussion section.

Reviewer #3: This is a very well done and well written study. I have a few minor corrections.

1. When the patient gets readmitted and is assigned the ICD category code of COVID 19 - does that mean he's got a re-infection, is PCR positive or has just recovered from COVID-19? It would be nice if there is one line to elaborate what that ICD code includes. (Page 7)

2. In page 10, under Readmission to patient care, line 10 - I think you meant to say 'Increased risk of readmission' and not 'mortality'.

3. The supplementary tables 8c and 9c give a lot of valuable information and I think it would be useful to include a more detailed description of those results in the discussion.

6. PLOS authors have the option to publish the peer review history of their article (what does this mean?). If published, this will include your full peer review and any attached files.

Reviewer #1: No

Reviewer #2: **Yes: **RAFAEL FERNANDEZ

Reviewer #3: **Yes: **Manisha Arthur

---

## [Author Response · Author response to Decision Letter 0]

9 Aug 2022

RESPONSE TO REVIEWERS IS ALSO INCLUDED IN THE REVISED COVER LETTER

Journal Requirements:

RESPONSE: This has been addressed. 

RESPONSE: Thank you, we intend to store the de-identified analysis data and analysis code on the following OSF site: https://osf.io/cg8ab/ once the manuscript is accepted for publication. We are still verifying levels of de-identified data which are appropriate to include but will have complete data uploaded upon acceptance. 

3. Please include your tables as part of your main manuscript and remove the individual files. Please note that supplementary tables (should remain/ be uploaded) as separate "supporting information" files

RESPONSE: Tables have been included in manuscript and supplementary tables have been uploaded as supporting information files. 

RESPONSE: Captions have been included. 

Reviewers' comments:

Reviewer's Responses to Questions

Comments to the Author

1. Is the manuscript technically sound, and do the data support the conclusions?

Reviewer #1: Partly

Reviewer #2: Yes

Reviewer #3: Yes

2. Has the statistical analysis been performed appropriately and rigorously? 

Reviewer #1: Yes

Reviewer #2: Yes

Reviewer #3: Yes

3. Have the authors made all data underlying the findings in their manuscript fully available?

Reviewer #1: No

Reviewer #2: Yes

Reviewer #3: Yes

4. Is the manuscript presented in an intelligible fashion and written in standard English?

Reviewer #1: Yes

Reviewer #2: Yes

Reviewer #3: Yes

5. Review Comments to the Author

Reviewer #1: Thank you for submitting your research article entitled “Mechanical ventilation for COVID-19: outcomes following discharge from inpatient treatment”. Authors evaluated the long-term outcomes of severe COVID-19 patients with or without mechanical ventilation (MV), its impacts on the hospital readmission, all-cause mortality, and reason for readmission. I think this manuscript is very meaningful for all medical personnel who are involved in COVID-19 pandemic. However, there are some major concerns that should be addressed by the authors at this time.

1. First of all, I think the topic of this study have little impact for the readers of this journal so that authors have not fully discussed in the DISCUSSION section. It is easy to imagine that the clinical prognosis of cases requiring mechanical ventilation is poor not only for other infections but also for COVID-19. Are there any differences in the characteristics or prognostic tendencies peculiar to patients with COVID-19?

RESPONSE: While we agree with the reviewer that some aspects of the current study reflect common sense findings (e.g. patients receiving mechanical ventilation have worse outcomes in follow-up) we disagree that the findings will have little impact on readers. We have clarified in the discussion (on page 28 of the revised manuscript) that though the association between MV treatment and adverse outcomes (such as readmission and mortality) shown in the current sample is comparable or less than other studies of MV for ARDS, the data we present is still extremely useful. This is because the mechanisms by which COVID-19 leads to respiratory distress may differ from traditional ARDS and because MV treatment for COVID-19 is not applied in the same manner as MV treatment for non-COVID-19 respiratory distress. Because the COVID-19 presents unique challenges to physicians treating respiratory distress, we feel the current results are worthy of adding to the literature. 

2. In the INTRODUCTION section, authors described that MV can produce lung injury, which lead to poor long-term outcomes in this condition. Among the present study populations, authors should discuss whether these poor outcomes and high readmission rates are due to the effects of mechanical ventilation management, the complications of COVID-19, or other factors?

RESPONSE: This is a critical distinction to make. Unfortunately given the available data, we believe that this analysis is beyond the scope of the current manuscript. Because knowledge about treating COVID-19 illness was evolving during the early course of the pandemic, it is difficult to identify whether the findings relate intrinsically to COVID-19 or are due to implementation of MV treatment. We have added a sentence to the limitations section on page 30 of the revised manuscript specifically articulating this stating: “Further it is difficult to identify ascertain the exact mechanisms of the association between MV and the adverse outcomes shown in this sample.; These associations could be it is possible that the associations are due to factors associated with COVID-19 illness, issues with the application of MV treatment, or some other unknown factor.” 

3. Authors described the limitation of data sampling in terms of readmission rate. Although I am not sure how many patients’ readmissions this healthcare system (Northwell Health) can cover, this limitation should be fully discussed since readmission rate is a primary outcome in this study. I would like to know how far the readmission rate in this study is from the real-world readmission rate.

RESPONSE: Thank you for this comment. We agree it is important to contextualize the extent to which the Northwell Health readmission rate is comparable to other local or national rates. To address this question, we considered the following information: (1) data collection during the early pandemic was varied across the country in terms of sample size, variable outcome type, and follow-up, resulting in a range of estimations of the “true” readmission rate nationally; (2) literature suggests that United States readmission rates were estimated to be between 4.5% and 19.19% We thus consider our readmission rate of 11.4% to be appropriately represent the New York area. 

4. Are there any standardized manual for MV management in these 23 medical facilities? If not, how did the physicians decide the indication of MV? Did you employ high-flow nasal cannula (HFNC) or non-invasive positive pressure ventilation (NPPV)? Moreover, are there any relationships between the duration of the intubation and readmission rate?

RESPONSE: The authors appreciate this comment and agree that a standardizing of practices for implementation ofing MV would aid in the interpretation of the findings. However, at the time of data collection, standards for COVID-19 treatment were still being developed, both across and within hospital systems. Given the variable nature of these decisions during the early pandemic, a standard is not available for this particular dataset. We’ve also clarified that MV guidelines for COVID-19 were developing during the early pandemic in revisions to the introduction on page 4 of the revised manuscript. In response to your second question, while we agree that examining the relation between duration and of intubation and readmission rate would provide important information. However, the goal of the current analysis was to compare patients treated with MV to a matched cohort who were not treated with MV. To eExamininge the association between ventilation duration and outcomes would require analysis of a different cohort and using a different design. While we do believe those analyses are important, we also believe it is beyond the scope of the current manuscript.

5. In the CONCLUSIONS, second paragraph [P15L11-P16L2] is not based on the data of this research. This is not conclusion of this study.

RESPONSE: We agree that this paragraph is not based on the data, and have therefore incorporated it into the discussion section rather than the conclusions. 

1. [P5L18] January 31, 2020 -> January 31, 2021

RESPONSE: Change applied. 

2. There are no “Table 1”, ”Table2” and “Table 3”.

RESPONSE: We have incorporated the tables into the body of the manuscript. 

3. For propensity score matching, please list the factors in order to make the adjustments.

RESPONSE: We agree that additional details are required to clarify which variables were used to match the MV and non-MV patients. We have revised the section discussing propensity scoring on pages 8 and 9 to include all details of which variables we utilized and the methods which that were utilized for the matching process. We have also expanded our description of confounding variables which were used in the matching process on pages 6 and 7 of the revised manuscript.

Reviewer #2: Butler et al. reported the outcome of COVID-19 patients following discharge differentiating between those who needed mechanical ventilation and those without it.

As expected, patients who required MV were more susceptible to readmission and lower survival. The authors state that this feature is clearly recognized in ordinary ICU patients (not COVID), so the objective of this study should be clarified. Was their hypothesis that COVID patients behave differently (better or worse) than ordinary ICU patients? Then, the readers should be informed about how different the present results are compared with series of non-COVID patients. Nevertheless, even these comparisons are tricky because non-COVID severe ARF commonly affect patients with severe sepsis or severe comorbidities that explained most of the long-term outcome worsening.

RESPONSE: The reviewer is correct about the goal of the study. Comparisons of MV treatment for COVID-19 and non-COVID-19 ARDS are difficult for many reasons. The most salient being that MV was not uniformly and rigorously applied for patients with severe COVID-19, especially early in the pandemic when hospital systems were overburdened and resources were scarce. We also agree with the reviewer that the mechanisms of outcomes following MV for ARDS may differ between COVID-19 and non-COVID-19 patients. As such, we have clarified our goals in the introduction. Our goal hope for this paper is to describe outcomes among patients who were treated with MV for COVID-19 and to highlight the need for additional follow-up and support among this population. 

Additional comments:

1.- Page 15, conclusions should be tailored by deleting the first 3 lines: “MV is an essential treatment ……... important”.

Also, the second paragraph about possible solutions must be moved to the end of the discussion section.

RESPONSE: Thank you, these changes have been applied. 

Reviewer #3: This is a very well done and well written study. I have a few minor corrections.

1. When the patient gets readmitted and is assigned the ICD category code of COVID 19 - does that mean he's got a re-infection, is PCR positive or has just recovered from COVID-19? It would be nice if there is one line to elaborate what that ICD code includes. (Page 7)

RESPONSE: Thank you for this comment. Three codes were included under the COVID-19 umbrella: U07.1 (confirmed diagnosis of COVID-19 documented by provider, a positive COVID-19 test, or a presumptive positive COVID-19 test) and J12.82 (pneumonia due to COVID-19). We’ve clarified that these diagnoses correspond to COVID-19 being a reason for readmission in the text. We have also clarified that Supplementary Table 1 shows all ICD-10 codes and how they relate to reasons for readmission. 

2. In page 10, under Readmission to patient care, line 10 - I think you meant to say 'Increased risk of readmission' and not 'mortality'.

RESPONSE: Thank you, yes, this change has been applied. 

3. The supplementary tables 8c and 9c give a lot of valuable information and I think it would be useful to include a more detailed description of those results in the discussion.

RESPONSE: We agree and have expanded our discussion of these sensitivity analyses in the results and the discussion sections of the paper. We have also clarified that these sensitivity analyses suggest the magnitude of the association between MV and outcomes may differ among sub-populations of patients with COVID-19 illness. We have also clarified noted that because of the small number of events in these sensitivity analyses, our results should be interpreted cautiously. 

6. PLOS authors have the option to publish the peer review history of their article (what does this mean?). If published, this will include your full peer review and any attached files.

Do you want your identity to be public for this peer review? For information about this choice, including consent withdrawal, please see our Privacy Policy.

Reviewer #1: No

Reviewer #2: Yes: RAFAEL FERNANDEZ

Reviewer #3: Yes: Manisha Arthur

---

## [Decision Letter · Decision Letter 1]

28 Oct 2022

Mechanical Ventilation for COVID-19: Outcomes Following Discharge from Inpatient Treatment

PONE-D-22-12638R1

Dear Dr. Butler,

We’re pleased to inform you that your manuscript has been judged scientifically suitable for publication and will be formally accepted for publication once it meets all outstanding technical requirements.

Kind regards,

Masaki Tago, M.D., Ph.D., FACP.

Academic Editor

PLOS ONE

Additional Editor Comments (optional):

Reviewers' comments:

Reviewer's Responses to Questions

**Comments to the Author**

1. If the authors have adequately addressed your comments raised in a previous round of review and you feel that this manuscript is now acceptable for publication, you may indicate that here to bypass the “Comments to the Author” section, enter your conflict of interest statement in the “Confidential to Editor” section, and submit your "Accept" recommendation.

Reviewer #1: All comments have been addressed

Reviewer #2: (No Response)

Reviewer #3: All comments have been addressed

2. Is the manuscript technically sound, and do the data support the conclusions?

Reviewer #1: Yes

Reviewer #2: Partly

Reviewer #3: Yes

3. Has the statistical analysis been performed appropriately and rigorously? 

Reviewer #1: Yes

Reviewer #2: Yes

Reviewer #3: Yes

4. Have the authors made all data underlying the findings in their manuscript fully available?

Reviewer #1: Yes

Reviewer #2: Yes

Reviewer #3: No

5. Is the manuscript presented in an intelligible fashion and written in standard English?

Reviewer #1: Yes

Reviewer #2: Yes

Reviewer #3: Yes

6. Review Comments to the Author

Reviewer #1: Thank you for resubmitting your updated research article entitled “Mechanical ventilation for COVID-19: outcomes following discharge from inpatient treatment”.

The authors responded appropriately to my questions and comments. I agree with your views and with the content of the revised paper. I believe that this research paper will be of high value not only to medical professionals working with COVID-19 pandemic, but also to epidemiological statisticians and the many citizens who need such information. Again, thank you for submitting your manuscript to this journal.

Reviewer #2: The authors have not answered my comments about a completely new reorientation. From my point of view MV Covid patients must be compared with MV NonCovid patients instead of nonventilated Covid.

In the present form, I think that the manuscript does not offer any new information.

Reviewer #3: All comments have been addressed.

7. PLOS authors have the option to publish the peer review history of their article (what does this mean?). If published, this will include your full peer review and any attached files.

Reviewer #1: No

Reviewer #2: No

Reviewer #3: No

---

## [Editor Report · Acceptance letter]

29 Dec 2022

PONE-D-22-12638R1 

Mechanical ventilation for COVID-19: Outcomes following discharge from inpatient treatment 

Dear Dr. Butler:

I'm pleased to inform you that your manuscript has been deemed suitable for publication in PLOS ONE. Congratulations! Your manuscript is now with our production department. 

Kind regards, 

on behalf of

Dr. Masaki Tago 

Academic Editor

PLOS ONE